# Association between malignancies and Marfan syndrome: a population-based, nested case–control study in Taiwan

Chin-Wang Hsu,[1,2] Jen-Chun Wang,[3,4] Wen-I Liao,[3] Wu-Chien Chien,[5,6] Chi-Hsiang Chung,[5,6,7] Chang-Huei Tsao,[5] Yung-Fu Wu,[5] Min-Tser Liao,[8] Shih-Hung Tsai[3]

W-CC and S-HT contributed equally.

For numbered affiliations see end of article.

**Correspondence to**
Dr. Wu-Chien Chien;
chienwu@mail.ndmctsgh.edu.tw and Dr Shih-Hung Tsai;
tsaishihung@yahoo.com.tw

## ABSTRACT

**Objective** Marfan syndrome (MFS) involves a deficiency of the structural extracellular matrix component fibrillin-1 and overactivation of the transforming growth factor-β (TGF-β) signalling pathway. The TGF-β signalling pathway also actively participates in malignant transformation. Although anecdotal case reports have suggested associations between MFS/MFS-like conditions and several haematological and solid malignancies, such associations have not been thoroughly evaluated in large-scale studies. We sought to use a nationwide healthcare insurance claim database to evaluate whether patients with MFS are at increased risk of malignancy.

**Patients and methods** We conducted a nested case–control analysis using a database extracted from Taiwan's National Health Insurance Research Database. All medical conditions for each case and control were categorised using the International Classification of Diseases, 9th Revision classifications. ORs and 95% CIs for associations between MFS and malignancies were estimated using conditional logistic regression and adjusted for comorbidities.

**Results** Our analyses included 1 153 137 cancer cases and 1 153 137 propensity score-matched controls. Relative to other subjects, patients with MFS had a significantly higher risk of having a malignancy (adjusted OR 3.991) and hypertension (adjusted OR 1.964) and were significantly more likely to be men. Malignancies originating from the head and neck and the urinary tract were significantly more frequent among patients with MFS than among subjects without MFS.

**Conclusion** Patients with MFS are at increased risk of developing various malignancies. Healthcare professionals should be aware of this risk when treating such patients, and increased cancer surveillance may be necessary for these patients.

## Strengths and limitations of this study

► The associations between Marfan syndrome (MFS)/MFS-like conditions and malignancies have not been thoroughly evaluated in large-scale studies. Thus, we used a nationwide healthcare insurance claim database to evaluate whether patients with MFS are at increased risk of malignancy. We found that patients with MFS are at increased risk of developing various malignancies.

► The National Health Insurance Research Database registry did not provide detailed information regarding laboratory results, family history and health-related lifestyle factors that could increase the risk of malignancy, and these factors represent potential confounding factors in this study.

► Our study can identify associations between MFS and malignancies, but a case–control study cannot prove a cause–effect relationship.

pathogenesis of this disease. Overactivated TGF-β signalling is associated with MFS and several MFS-like conditions, including Loeys-Dietz syndrome, Shprintzen-Goldberg syndrome, aneurysm–osteoarthritis syndrome and syndromic thoracic aortic aneurysm. These conditions are also clearly associated with degenerative non-inflammatory structural cardiovascular diseases, including aortic root dilatation, thoracic aneurysm and aortic dissection.[1 2] The TGF-β signalling pathway also actively participates in malignant transformation. In tumour cells, TGF-β loses its antiproliferative response and becomes an oncogenic factor; as a result, TGF-β function is impaired in various solid and haematological malignancies.[3] TGF-β-induced epithelial–mesenchymal transition and reversion from mesenchymal to epithelial phenotypes contribute to the survival and dissemination of malignant cells.[4] Anecdotal case reports have suggested an association between MFS/MFS-like conditions and several haematological and solid malignancies.[5–15] However,

## INTRODUCTION

Marfan syndrome (MFS) is a pleiotropic connective tissue disease caused by a deficiency of the structural extracellular matrix component fibrillin-1 (FBN-1). The study of murine models of MFS has revealed the involvement of the transforming growth factor-β (TGF-β) signalling pathway in the

associations between MFS/MFS-like conditions and malignancies have not been thoroughly evaluated in large-scale studies. In this respect, we sought to use a nationwide healthcare insurance claim database to evaluate whether patients with MFS are at increased risk of malignancy.

## METHODS
### Data source
Data for our nationwide, population-based, nested case–control study were obtained from inpatient care records and registration files from the Taiwan National Health Insurance Research Database (NHIRD). The National Health Insurance programme was implemented in 1995 and provides healthcare coverage to 99% of the Taiwanese population (more than 23 million people). The accuracy of the NHIRD with respect to diagnoses for major diseases, such as stroke and acute coronary syndrome, has been validated.[16 17] The confidentiality of individuals was protected by using encrypted personal identification to avoid the possibility of ethical violations related to the study data. This investigation was conducted in accordance with the Declaration of Helsinki and other relevant guidelines. This study was approved by the Institutional Review Board of Tri-Service General Hospital, National Defense Medical Center, Taipei, Taiwan (TSGH IRB number B-104-21).

### Cancer cases and controls
This study involved a nested case–control design. Using the NHIRD, we selected adult patients >18 years of age who had been diagnosed with a malignancy based on the International Classification of Diseases, 9th Revision, Clinical Modification (ICD-9-CM) codes (140–208) between 2000 and 2013 and confirmed these patients' diagnoses by linking them to cases registered in the Catastrophic Illness Patient Database. The date of the first malignancy diagnosis was defined as the index date. We identified patients with MFS using the ICD-9-CM code 759.82. A sample of control candidates was selected for comparison from individuals in the NHIRD who were without malignancies. Patients in the study and control groups were selected via 1:1 matching by age, sex, number of medical follow-ups and comorbidities, including hypertension (ICD-9-CM 401–405), diabetes (ICD-9-CM 250), hyperlipidaemia (ICD-9-CM 272.0–272.4), chronic obstructive pulmonary disease (COPD) (ICD-9-CM 490–496), alcoholism (ICD-9-CM 303) and obesity (ICD-9-CM 278). Data on the use of angiotensin-converting enzyme inhibitors and angiotensin receptor blockers were acquired from the Longitudinal Health Insurance Database 2005, a subdatabase of the NHIRD. All insurance claims were scrutinised by medical reimbursement specialists, and peer review was undertaken according to standard and clinical diagnostic criteria, such as the pulmonary function test for COPD. Therefore, the diagnoses of COPD in this study should be highly reliable. Both the NHIRD and

catastrophic illness certificate have been well-validated internally and externally in several studies.[16 18 19]

### Outcome measurements
Only patients diagnosed with MFS prior to the index date were considered. MFS was identified from the NHIRD by using the corresponding ICD-9 code (ICD-9-CM code 759.82). The following covariates were included: age, gender, hypertension, diabetes mellitus (DM), hyperlipidaemia, COPD, alcoholism and obesity.

### Statistical analysis
Patients' clinical characteristics were expressed in numerical form. Categorical variables, which were presented as percentages, were compared using $X^2$ tests and Fisher's exact test. Continuous variables, which were presented as the mean and SD, were compared using Student's t-tests. The primary goal of this study was to determine whether a patient's clinical characteristics, such as MFS, were associated with malignancies. Associations between those outcomes (prognoses) and clinical characteristics were investigated using conditional logistic regression in a generalised estimating equation (GEE) model. The regression results are presented as adjusted ORs with corresponding 95% CIs. The threshold for statistical significance was p<0.05. All data analyses were conducted using SPSS V.18 (SPSS).

## RESULTS
A flow diagram of the patient enrolment scheme is presented in figure 1. During the study period, a total of 1 153 137 patients in follow-up after being diagnosed with a malignancy were identified in the NHIRD, which included a total of 13 139 306 inpatients. Other patients from the same database who did not have malignancies and were matched by age, gender, number of medical follow-ups and comorbidities were designated controls. As indicated in table 1, as expected, there were no statistically significant differences between the case and control groups with respect to gender, age, number of medical follow-ups and comorbidities, including hypertension, DM, hyperlipidaemia, COPD, alcoholism and obesity, after matching.

At the end of the 13-year study period, the incidence of MFS was significantly higher among patients with malignancies compared with patients without malignancies over the corresponding observation period (0.008% vs 0.003%, p<0.001). Among patients with MFS, there were no statistically significant differences regarding the use of angiotensin-converting enzyme inhibitors and angiotensin receptor blockers between patients with or without malignancies (20.93% vs 21.33%, p=0.95; 18.60% vs 19.11%, p=0.93, respectively). Table 2 indicates that malignancy (adjusted OR 3.991, 95% CI 2.555 to 6.235, p<0.001), maleness (adjusted OR 2.103, 95% CI 1.427 to 3.098, p<0.001) and hypertension (adjusted OR 1.964,

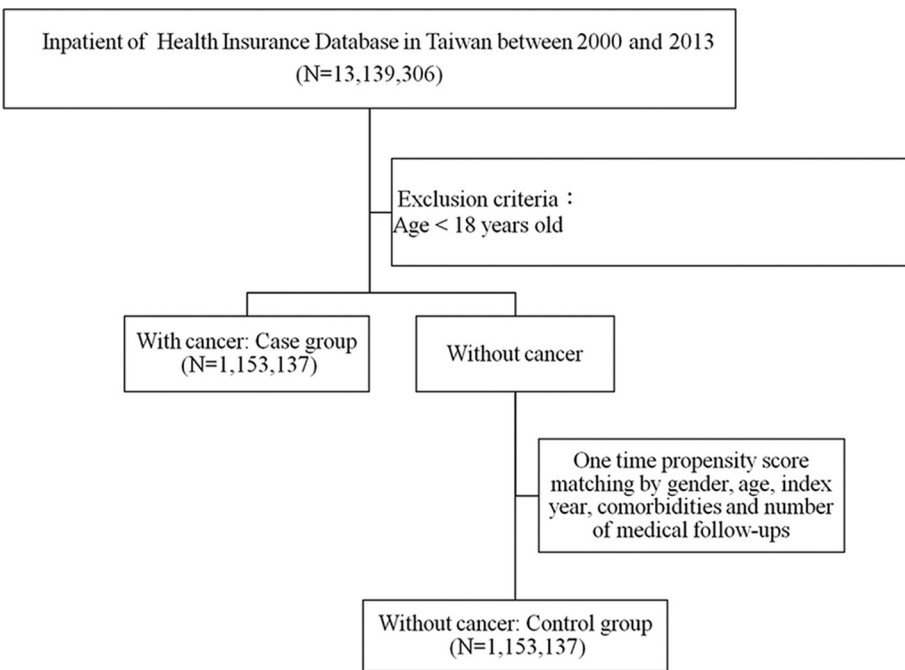

**Figure 1** Patient selection flow chart.

95% CI 1.258 to 3.064, p=0.013) were significantly associated with patients with MFS.

As indicated in table 3, malignancies originating from the head and neck (adjusted OR 8.714, 95% CI 2.477 to 30.649, p<0.001), oesophagus (adjusted OR 0.970, 95% CI 3.943 to 30.522, p<0.001), stomach (adjusted OR 4.639, 95% CI 1.743 to 12.345, p=0.002), colon and rectum (adjusted OR 3.134, 95% CI 1.370 to 7.169, p=0.007), liver (adjusted OR 3.049, 95% CI 1.349 to 6.980, p=0.007), female genital organs (adjusted OR 4.055, 95% CI 1.417 to 11.606, p=0.009), prostate (adjusted OR 6.423,

95% CI 2.355 to 17.517, p<0.001), urinary tract (adjusted OR 6.478, 95% CI 2.974 to 14.112, p<0.001) and thyroid (adjusted OR 10.485, 95% CI 4.251 to 25.861, p<0.001) and haematological malignancies (adjusted OR 6.981, 95% CI 2.800 to 17.404, p<0.001) were significantly more frequent among patients with MFS than among subjects without MFS.

As indicated in table 4, patients with MFS acquired malignancies at a younger age than those without MFS. During the study period, patients with MFS had a higher mortality rate and died younger than those without MFS.

| Table 1 | Baseline characteristics of the study population | | |
|---|---|---|---|
| | **Patients with malignancy (n=1 153 137)** | **Patients without malignancy (n=1 153 137)** | **p Value** |
| MFS | 93 (0.008%) | 27 (0.002%) | <0.001 |
| Gender | | | 0.999 |
| Male | 657 188 (56.990%) | 657 188 (56.990%) | |
| Female | 495 969 (43.010%) | 495 969 (43.010%) | |
| Age (years) | 63.52±15.21 | 63.49±18.42 | 0.178 |
| Comorbidities | | | |
| Hypertension | 150 555 (13.056%) | 150 512 (13.052%) | 0.933 |
| Diabetes | 127 884 (11.090%) | 127 894 (11.091%) | 0.978 |
| Hyperlipidaemia | 7235 (0.627%) | 7233 (0.627%) | 0.987 |
| COPD | 48 668 (4.220%) | 48 656 (4.219%) | 0.969 |
| Alcoholism | 1052 (0.091%) | 1042 (0.090%) | 0.827 |
| Obesity | 230 (0.020%) | 247 (0.021%) | 0.436 |
| Medical follow-ups | 4.84±10.70 | 4.82±10.87 | 0.159 |

COPD, chronic obstructive pulmonary disease; MFS, Marfan syndrome.

**Table 2** Factors of MFS determined using conditional logistic regression in a GEE model

|  | Crude OR | 95% CI | p Value | Adjusted OR* | 95% CI | p Value |
|---|---|---|---|---|---|---|
| **Malignancy** | | | | | | |
| Without | Reference | | | Reference | | |
| With | 3.444 | 2.012 to 4.671 | <0.001 | 3.999 | 2.578 to 6.301 | <0.001 |
| **Gender** | | | | | | |
| Male | 2.187 | 1.490 to 3.211 | <0.001 | 2.110 | 1.429 to 3.101 | <0.001 |
| Female | Reference | | | Reference | | |
| **Hypertension** | | | | | | |
| Without | Reference | | | Reference | | |
| With | 1.503 | 1.013 to 2.324 | 0.001 | 1.962 | 1.224 to 3.000 | 0.010 |
| **Diabetes** | | | | | | |
| Without | Reference | | | Reference | | |
| With | 0.993 | 0.546 to 1.717 | 0.785 | 0.963 | 0.527 to 1.688 | 0.809 |
| **Hyperlipidaemia** | | | | | | |
| Without | Reference | | | Reference | | |
| With | 0.748 | 0.197 to 2.862 | 0.704 | 0.711 | 0.214 to 3.201 | 0.764 |
| **COPD** | | | | | | |
| Without | Reference | | | Reference | | |
| With | 0.299 | 0.025 to 1.374 | 0.102 | 0.398 | 0.027 to 1.401 | 0.083 |
| **Alcoholism** | | | | | | |
| Without | Reference | | | Reference | | |
| With | 0.000 | – | 0.989 | 0.000 | – | 0.990 |
| **Obesity** | | | | | | |
| Without | Reference | | | Reference | | |
| With | 1.014 | 0.859 to 1.086 | 0.177 | 1.010 | 0.803 to 1.020 | 0.243 |

Adjusted by variables listed in the table.

COPD, chronic obstructive pulmonary disease; GEE, generalised estimating equation; MFS, Marfan syndrome.

## DISCUSSION

Using a nationwide population database, we have demonstrated for the first time that MFS is associated with the development of certain malignancies. No systemic approaches have previously been used to elucidate associations between MFS and malignancies.

Numerous case reports have addressed associations between MFS and various solid malignancies, including thyroid cancer,[8] osteosarcoma,[9] mesothelioma,[14] oesophageal cancer,[20] angiosarcoma,[21] testicular cancer[22] and Wilms' tumour.[23] Multiple endocrine neoplasia syndromes that are characterised by benign and malignant changes in multiple endocrine organs and incidental changes in nervous, muscular and connective tissues can also manifest marfanoid features.[24 25] In our study, we observed that the frequencies of head and neck and urinary tract malignancies were significantly increased in patients with MFS. Accumulated case reports have also indicated that MFS and MFS-like conditions, such as Loeys-Dietz syndrome and Ehlers-Danlos syndrome (EDS), might be associated with various haematological malignancies and non-Hodgkin's lymphoma.[5–7 10–13 15 26–28] Larger, international studies should be conducted to study these potential associations.

As integral components of microfibrils, which provide strength and elasticity to the extracellular matrix, fibrillins, particularly FBN-1 and FBN-2, are thought to be involved in cancer pathogenesis and maintenance of the pluripotency of embryonic stem cells. Fibrillins are important for controlling the growth and differentiation of cells that they surround via interaction with integrins and growth factors as well as regulation of members of the TGF-β superfamily.[29] An FBN-1 deficiency impairs targeting of the large latent complex by the extracellular matrix, resulting in the unrestrained release of TGF-β ligands. An elevated TGF-β level in patients with MFS is correlated with larger aortic root diameters and faster aortic root growth.[30] Dysregulated TGF-β2 is also associated with vascular EDS.[31] TGF-β actively participates in malignant transformation and progression. Evidence has revealed associations between TGF-β and various solid malignancies, including head and neck,[32] bladder,[33] prostate,[34] colon,[35] lung,[36] breast,[37] liver[38] and renal cell cancer.[39] Mutations in TGFBR2, a putative tumour

**Table 3** Analysis of malignancy subgroup using conditional logistic regression in a GEE model in patients with or without MFS

| Malignancy (with vs without) | Adjusted OR | 95% CI | p Value |
|---|---|---|---|
| Total | 3.999 | 2.578 to 6.301 | <0.001 |
| Head and neck | 8.714 | 2.477 to 30.649 | <0.001 |
| Oesophagus | 10.970 | 3.943 to 30.522 | <0.001 |
| Stomach | 4.639 | 1.743 to 12.345 | 0.002 |
| Colon and rectum | 3.134 | 1.370 to 7.169 | 0.007 |
| Liver | 3.049 | 1.349 to 6.980 | 0.007 |
| Trachea, bronchus and lung | 1.938 | 0.705 to 5.329 | 0.200 |
| Female breast | 0.477 | 0.060 to 3.782 | 0.483 |
| Female genital organs | 4.055 | 1.417 to 11.606 | 0.009 |
| Prostate | 6.423 | 2.355 to 17.517 | <0.001 |
| Bladder, ureter and kidney | 6.478 | 2.974 to 14.112 | <0.001 |
| Thyroid | 10.485 | 4.251 to 25.861 | <0.001 |
| Haematopoietic malignancy | 6.981 | 2.800 to 17.404 | <0.001 |
| Others | 2.900 | 1.434 to 5.863 | 0.003 |

Adjusted OR: adjusted for variables listed in table 2.
GEE, generalised estimating equation; MFS, Marfan syndrome.

suppressor gene implicated in several malignancies, are also associated with inherited connective tissue disorders.[40] The TGF-β/SMAD signalling pathway is constitutively activated in natural killer cells in patients with acute lymphoblastic leukaemia but not in healthy controls.[41] Functional variants of TGF-β1 genes may be significantly associated with the aetiopathogenesis of acute myeloid leukaemia.[42] TGF-β1 also induces the PI3K/Akt/NF-κB signalling pathway during the recruitment of malignant cells in chronic myeloid leukaemia.[43] A previous study performed in our laboratory has confirmed the association between malignancies and aortic aneurysms (unpublished data).

## LIMITATIONS

The strength of our study is its population-based database design. We identified malignancies using the dual approaches of assessing ICD-9 code records and searching the registry of the Catastrophic Illness Patient Database to increase the accuracy of our data. We excluded

**Table 4** Analysis of age and survival regarding malignancy in patients with or without MFS

| MFS | With (%) | Without (%) | p Value |
|---|---|---|---|
| Age at first inpatient visit due to malignancy | 52.74±16.92 | 58.52±15.29 | <0.001 |
| Age at death during the follow-up period | 67.35±1.63 | 70.84±11.85 | <0.001 |
| Patient mortality during the follow-up period | 2.15% | 0.197% | <0.001 |

MFS, Marfan syndrome.

confounding factors of malignancy, including comorbidities. Although we extensively adjusted our results by using multivariate logistic regression models, our study nonetheless exhibited several limitations and did not address certain confounders. First, MFS might be underdiagnosed in Asian populations.[44] MFS is diagnosed using the Ghent criteria, which are primarily based on clinical features of Caucasian MFS populations; however, clinical features of the cardiovascular, ocular and skeletal systems significantly differ between Caucasian and Asian MFS populations.[44–46] Second, the NHIRD registry could not provide detailed information regarding laboratory results, family histories and health-related lifestyle factors, such as alcohol consumption or tobacco use, that can increase the risk of malignancy and were potential confounding factors in this study. In our investigation, we also considered COPD incidence as a proxy variable for tobacco use to diminish the potential confounding effect of tobacco use on our results.[47] Third, this investigation was designed as a case–control study because this approach is efficient for relatively rare diseases or diseases with a long latency period, such as MFS, due to cost-related and time-related considerations. Our study can identify associations between MFS and malignancies, but a case–control study cannot prove a cause–effect relationship. Ethnic differences in the association between MFS and malignancy should be further explored in larger, international, prospective follow-up studies enrolling multiple ethnic populations.

## CONCLUSIONS

Patients with MFS are at increased risk of developing various malignancies. Healthcare professionals should be aware of this risk when treating patients with MFS. Relative to other patients, patients with MFS may require additional cancer surveillance.

**Author affiliations**
[1]Department of Emergency Medicine, School of Medicine, College of Medicine, Taipei Medical University, Taipei, Taiwan
[2]Department of Emergency and Critical Medicine, Wan Fang Hospital, Taipei Medical University, Taipei, Taiwan
[3]Department of Emergency Medicine, Tri-Service General Hospital, National Defense Medical Center, Taipei, Taiwan
[4]Institute of Clinical Medicine, National Yang-Ming University, Taipei, Taiwan
[5]Department of Medical Research, Tri-Service General Hospital, National Defense Medical Center, Taipei, Taiwan
[6]School of Public Health, National Defense Medical Center, Taipei, Taiwan
[7]Taiwanese Injury Prevention and Safety Promotion Association, Taoyuan, Taiwan
[8]Department of Pediatrics, Taoyuan Armed Forces General Hospital, Taoyuan, Taiwan

**Contributors** C-WH and M-TL conceptualised and designed the study. J-CW, W-IL and S-HT reviewed relevant literature and drafted the manuscript. W-CC provided the dataset and coordinated the study. C-HC, C-HT and Y-FW analysed the data. All authors collected and interpreted the data and approved the manuscript.

**Funding** This study was supported by grants from the Tri-Service General Hospital, National Defense Medical Center, Taipei, Taiwan (TSGH-C106-002 and TSGH-C106-048); Taoyuan Armed Forces General Hospital, Taoyuan, Taiwan (10514) and Ministry of Science and Technology (MOST 106-2314-B-016-031-.

**Competing interests** None declared.

**Provenance and peer review** Not commissioned; externally peer reviewed.

**Data sharing statement** No additional data are available.

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
