## [Reviewer comments · BMJ Open]

ARTICLE DETAILS

TITLE (PROVISIONAL)	The association between malignancies and Marfan syndrome: a population-based, nested case-control study in Taiwan
AUTHORS	Hsu, Chin-Wang; Wang, Jen-Chun; Liao, Wen-I; Chien, Wu-Chien; Chung, Chi-Hsiang; Tsao, Chang-Huei; Wu, Yung-Fu; Liao, Min-Tser; Tsai, Shih-Hung

VERSION 1 – REVIEW

REVIEWER	Reed Pyeritz University of Pennsylvania Philadelphia, PA, USA
REVIEW RETURNED	15-May-2017

GENERAL COMMENTS	The hypothesis is interesting but the approach and available clinical data are not sufficient to answer the questions. First, if enough forms of neoplasia are examined statistically, then by chance alone one or two will achieve a p values < 0.05. Second, the fact that Asians with Marfan syndrome differ phenotypically from Caucasians needs to be considered more substantively (Franken et al, Circulation Journal 2013;77:2793-8). Third, the age differences among the Marfan subjects and those with neoplasia need to be considered. Fourth, it isn't clear to me why you 'excluded patients who had been diagnosed with MFS within 6 months prior to being diagnosed with a new malignancy. Fifth, the use of medications by the MFS cohort, especially an angiotensin receptor blocker (which would complicate the TGF-beta hypothesis) should have been addressed in the data retrieval. Sixth, for those diagnosed with MFS, they are more likely to be in routine medical follow-up than an average person of the same age, and therefore more likely to have a malignancy diagnosed. Seventh, on page 12, the TGF-beta hypothesis is mentioned in regard to Ehlers-Danlos syndrome, none of the varieties of which involve the TGF-beta pathways. Eighth, it is mentioned that (p. 15) that COPD was used as a proxy for tobacco use; in fact people with MFS much more frequently have pulmonary cysts, so I wonder how COPD was being evaluated. Finally, there was just such a small number of malignancies found in the MFS subjects that I return to validity of the statistical methods.
---

REVIEWER	Hazem Elsebaie Cairo University - Egypt
REVIEW RETURNED	09-Jun-2017

GENERAL COMMENTS	Interesting subject, great effort, rich data and well organized manuscript. Some points requiring the author's comment: 1. Page 8: Patients in the study and control groups were selected via 1:1 matching by age, sex, and comorbidities, including hypertension (ICD-9-CM 401-405), diabetes (ICD-9-CM 250), hyperlipidemia (ICD-9-CM 272.0-272.4), chronic obstructive pulmonary disease (COPD) (ICD-9-CM 490-496), alcoholism (ICD-9-CM 303), and obesity (ICD-9-CM 278). Page 10 As indicated in Table 1, as expected, there were no statistically significant differences between the case and control groups with respect to gender, age and co-morbidities, including hypertension, DM, hyperlipidemia, COPD, alcoholism and However: Page 4: NHIRD registry could not provide detailed information regarding laboratory results, family histories and health-related lifestyle factors, that could increase the risk of malignancy and were potential confounding factors in this study. Page 14: Second, the NHIRD registry could not provide detailed information regarding laboratory results, family histories and health-related lifestyle factors, such as alcohol consumption or tobacco use,.... The authors need to clarify: does the NHIRD registry provide information regarding health-related lifestyle factors such as alcohol consumption or tobacco or not? If not then what is the code of alcoholism (ICD-9-CM 303)? It is kind of surprising to find a up to date health registry not recording such a relevant health related aspect as tobacco use. 2. Page 8: The date of the first malignancy diagnosis was defined as the index date. We excluded patients who had been diagnosed with Marfan syndrome (ICD-9-CM 759.82) within 6 months prior to being diagnosed with a new malignancy. Can the authors explain the reason for this exclusion as long as it is prior to the index date?
---

REVIEWER	Giuseppe Biondi Zoccai Sapienza University of Rome, Italy
REVIEW RETURNED	12-Jul-2017

GENERAL COMMENTS	The authors report an interesting case-control study appraising the association between cancer and Marfan syndrome, suggesting that this condition is associated with an increased risk of cancer (in particular of some types of cancer). Despite the work strengths, I recommend addressing the following comments: 1. Compare with interaction testing the different odds ratios for cancer types in the Marfan group and in the control group. 2. Appraise time to cancer and/or cancer death with survival analysis.
--

REVIEWER	Xiaoju Zhang Abbott Laboratories, USA No competing interest.
REVIEW RETURNED	20-Jul-2017

GENERAL COMMENTS	Taking advantage of huge size of population from million level Taiwan's NHIRD database, the authors of this manuscript conducted an association study between Marfan syndrome (MFS) with multiple previous anecdotally reported MFS associated malignancies, including head, neck, urinary tract, hematological origin, gastrointestinal tract, female genital tract, and thyroid malignancies. Previous studies are addressed well to support the motivation of the described work. Besides the large size samples, the manuscript describes clearly about the statistics, and the statistical reports follow the proper practice, for example P value below the cutoff is recorded as <0.001 instead of the exact number. Both positive and negative findings added values to the clinical field. At the same time, the authors discussed thoroughly about the potential limitations of this study. In total, this manuscript holds a high quality.
---

VERSION 1 – AUTHOR RESPONSE

Reviewer: 1

Reviewer Name: Reed Pyeritz

Institution and Country: University of Pennsylvania, Philadelphia, PA, USA

Please state any competing interests or state 'None declared': None declared

Please leave your comments for the authors below

Comment: The hypothesis is interesting but the approach and available clinical data are not sufficient to answer the questions.

First, if enough forms of neoplasia are examined statistically, then by chance alone one or two will achieve a p values < 0.05.

Response: Thanks for your valuable comments. We conducted this study because of the TGF-β hypothesis and because numerous case reports have highlighted the association between MFS and malignancies. We emphasize that this is a hypothesis-driven, population-based case-control study. The p value for having MFS among those with a malignancy was <0.0001 in this case-control design, although the number was limited. The sampling error would be minimized in this setting. Appropriate matching procedures could further increase the validity of the study. Please see page 15, lines 7-12; page 13; and the limitations section on page 18, lines 4-10.

Comment: Second, the fact that Asians with Marfan syndrome differ phenotypically from Caucasians needs to be considered more substantively (Franken et al, Circulation Journal 2013;77:2793-8).

Response: We agree with your comment that phenotypic differences in MFS between Asian and Caucasian populations should be considered. Therefore, we mentioned this difference in the limitations section. Please see page 17, lines 13-17, and page 18, lines 8-10.

Comment: Third, the age differences among the Marfan subjects and those with neoplasia need to be considered.

Response: Thanks for your valuable comment. The control group was matched by age and gender before performing any comparisons. Since MFS is a hereditary disease, we further analyzed the age at cancer diagnosis. We found that patients with MFS acquired malignancies at a younger age than those without MFS. In addition, patients with MFS had a higher mortality rate and died younger than those without MFS. Please see page 13, lines 10-12, and the new Table 4.

Comment: Fourth, it isn't clear to me why you 'excluded patients who had been diagnosed with MFS within 6 months prior to being diagnosed with a new malignancy.

Response: Thanks for your valuable comment. Since MFS is a hereditary disease, the wash-out period might not be needed under these circumstances. Therefore, we have reanalyzed the dataset. We removed this exclusion criterion and found no change in the number of cases. Please see revised Table 1 and Figure 1.

Comment: Fifth, the use of medications by the MFS cohort, especially an angiotensin receptor blocker (which would complicate the TGF-beta hypothesis) should have been addressed in the data retrieval.

Response: Thanks for your valuable comment. We have reevaluated the data and found that among patients with MFS, there were no statistically significant differences regarding the use of angiotensin-converting enzyme inhibitors and angiotensin receptor blockers between patients with or without malignancies (20.93% vs 21.33, $p=0.95$; 18.60% vs 19.11%, $p=0.93$, respectively). Please see page 11, lines 5-8.

Comment: Sixth, for those diagnosed with MFS, they are more likely to be in routine medical follow-up than an average person of the same age, and therefore more likely to have a malignancy diagnosed.

Response: We agree with your comments. We have reanalyzed our data. In addition to age, gender and comorbidities, we further matched the case and control groups based on the number of medical follow-ups. As shown in revised Table 3, there was no difference regarding the number of medical follow-ups. Please see page 8, line 6; page 10, lines 8-9; revised Table 1; and Figure 1.

Comment: Seventh, on page 12, the TGF-beta hypothesis is mentioned in regard to Ehlers-Danlos syndrome, none of the varieties of which involve the TGF-beta pathways.

Response: Thanks for your comments. We have added some relevant information in the revised discussion section. Please see page 16, line 10.

Comment: Eighth, it is mentioned that (p. 15) that COPD was used as a proxy for tobacco use; in fact people with MFS much more frequently have pulmonary cysts, so I wonder how COPD was being evaluated.

Response: We agree with your comments. All insurance claims were scrutinized by medical reimbursement specialists, and peer review was undertaken according to standard and clinical diagnostic criteria, such as the pulmonary function test for COPD. Therefore, the diagnoses of COPD in this study should be highly reliable. Both the NHIRD and the catastrophic illness certificate have been well-validated internally and externally in several studies. 1-3 Please see page 8, lines 12-17.

Comment: Finally, there was just such a small number of malignancies found in the MFS subjects that I return to validity of the statistical methods.

Response: Thanks for your comment. This investigation was designed as a case-control study because this approach is efficient for relatively rare diseases or diseases with a long latency period, such as MFS, due to cost- and time-related considerations. Our study can identify associations between MFS and malignancies, but a case-control study cannot prove a cause-effect relationship. Ethnic differences in the association between MFS and malignancy should be further explored in larger, international, prospective follow-up studies enrolling multiple ethnic populations. Please see the limitations section, page 18, lines 3-10.

Reviewer: 2

Reviewer Name: Hazem Elsebaie

Institution and Country: Cairo University - Egypt

Please state any competing interests or state 'None declared': None

Please leave your comments for the authors below
Interesting subject, great effort, rich data and well organized manuscript.
Some points requiring the author's comment:

Comment 1. Page 8:

Patients in the study and control groups were selected via 1:1 matching by age, sex, and comorbidities, including hypertension (ICD-9-CM 401-405), diabetes (ICD-9-CM 250), hyperlipidemia (ICD-9-CM 272.0-272.4), chronic obstructive pulmonary disease (COPD) (ICD-9-CM 490-496), alcoholism (ICD-9-CM 303), and obesity (ICD-9-CM 278).

Comment: Page 10

As indicated in Table 1, as expected, there were no statistically significant differences between the case and control groups with respect to gender, age and co-morbidities, including hypertension, DM, hyperlipidemia, COPD, alcoholism and

However:

Comment

Page 4:

NHIRD registry could not provide detailed information regarding laboratory results, family histories and health-related lifestyle factors, that could increase the risk of malignancy and were potential confounding factors in this study.

Comment: Page 14

Second, the NHIRD registry could not provide detailed information regarding laboratory results, family histories and health-related lifestyle factors, such as alcohol consumption or tobacco use,....

The authors need to clarify: does the NHIRD registry provide information regarding health-related lifestyle factors such as alcohol consumption or tobacco or not? If not then what is the code of alcoholism (ICD-9-CM 303)? It is kind of surprising to find a up to date health registry not recording such a relevant health related aspect as tobacco use.

Response: Thanks for your comments. The NHIRD is a healthcare insurance claim database based on ICD coding for diagnosis and treatment. Data were not directly acquired from patients; therefore, health-related lifestyle factors are not included in the database. However, patients who have a negative effect on their quality of life associated with alcohol usage would have been coded with alcoholism accordingly. Previous studies have used COPD as a surrogate for tobacco use.⁴ Please see page 8, lines 12-17, in the revised limitations section.

Comment 2. Page 8:

The date of the first malignancy diagnosis was defined as the index date. We excluded patients who had been diagnosed with Marfan syndrome (ICD-9-CM 759.82) within 6 months prior to being diagnosed with a new malignancy.

Can the authors explain the reason for this exclusion as long as it is prior to the index date?

Response: Thanks for your comments. Since MFS is a hereditary disease, the wash-out period might not be needed under these circumstances. Therefore, we have reanalyzed the data. We removed this exclusion criterion and found no change in the number of cases. Please see revised Table 1 and Figure 1.

Reviewer: 3

Reviewer Name: Giuseppe Biondi Zoccai

Institution and Country: Sapienza University of Rome, Italy

Please state any competing interests or state 'None declared': None declared

Please leave your comments for the authors below

The authors report an interesting case-control study appraising the association between cancer and Marfan syndrome, suggesting that this condition is associated with an increased risk of cancer (in particular of some types of cancer). Despite the work strengths, I recommend addressing the following comments:

Comment 1. Compare with interaction testing the different odds ratios for cancer types in the Marfan group and in the control group.

Response: Thanks for your comments. We analyzed the data as suggested. As shown in revised Table 3, malignancies originating from the head and neck (adjusted OR=8.714, 95% CI: 2.477-30.649, $p<0.001$), esophagus (adjusted OR=10.970, 95% CI: 3.943-30.522, $p<0.001$), stomach (adjusted OR=4.639, 95% CI: 1.743-12.345, $p=0.002$), colon and rectum (adjusted OR=3.134, 95% CI: 1.370-7.169, $p=0.007$), liver (adjusted OR=3.049, 95% CI: 1.349-6.980, $p=0.007$), female genital organs (adjusted OR=4.055, 95% CI: 1.417-11.606, $p=0.009$), prostate (adjusted OR=6.423, 95% CI: 2.355-17.517, $p<0.001$), urinary tract (adjusted OR=6.478, 95% CI: 2.974-14.112, $p<0.001$), and thyroid (adjusted OR=10.485, 95% CI: 4.251-25.861, $p<0.001$) and hematological malignancies (adjusted OR=6.981, 95% CI: 2.800-17.404, $p<0.001$) were significantly more frequent among patients with MFS than among subjects without MFS. Please see page 12, lines 3-11, and revised Table 3.

Comment 2. Appraise time to cancer and/or cancer death with survival analysis.

Response: Thanks for your valuable comment. Since MFS is a hereditary disease, we further analyzed the age at diagnosis of any type of cancer. We found that patients with MFS acquired malignancies at a younger age than those without MFS. In addition, patients with MFS had a higher mortality rate and died younger than those without MFS. Please see page 13, lines 10-12, and revised Table 4.

Reviewer: 4

Reviewer Name: Xiaoju Zhang

Institution and Country: Abbott Laboratories, USA

Please state any competing interests or state 'None declared': No competing interest.

Please leave your comments for the authors below

Comment: Taking advantage of huge size of population from million level Taiwan's NHIRD database, the authors of this manuscript conducted an association study between Marfan syndrome (MFS) with multiple previous anecdotally reported MFS associated malignancies, including head, neck, urinary tract, hematological origin, gastrointestinal tract, female genital tract, and thyroid malignancies. Previous studies are addressed well to support the motivation of the described work. Besides the large size samples, the manuscript describes clearly about the statistics, and the statistical reports follow the proper practice, for example P value below the cutoff is recorded as <0.001 instead of the exact number. Both positive and negative findings added values to the clinical field. At the same time, the authors discussed thoroughly about the potential limitations of this study. In total, this manuscript holds a high quality.

Response: Thanks for your valuable comments.

References

1. Lee LJ-H, Chang Y-Y, Liou S-H, Wang J-D. Estimation of benefit of prevention of occupational cancer for comparative risk assessment: methods and examples. Occupational and Environmental Medicine. 2012;69(8):582-586.
2. Cheng CL, Kao YH, Lin SJ, Lee CH, Lai ML. Validation of the National Health Insurance Research Database with ischemic stroke cases in Taiwan. Pharmacoepidemiology and drug safety. 2011;20(3):236-242.
3. Wu CY, Chan FK, Wu MS, et al. Histamine2-receptor antagonists are an alternative to proton pump inhibitor in patients receiving clopidogrel. Gastroenterology. 2010;139(4):1165-1171.
4. Yu TM, Chuang YW, Yu MC, et al. Risk of cancer in patients with polycystic kidney disease: a propensity-score matched analysis of a nationwide, population-based cohort study. The Lancet Oncology. 2016.

VERSION 2 – REVIEW

REVIEWER	Hazem Elsebaie Cairo University, Egypt
REVIEW RETURNED	27-Aug-2017

GENERAL COMMENTS	The author's reply, clarifications and amendments seem satisfactory.
--

REVIEWER	Giuseppe Biondi-Zoccai Sapienza University of Rome, Latina, Italy
REVIEW RETURNED	15-Aug-2017

GENERAL COMMENTS	All my comments have been satisfactorily addressed.
---

REVIEWER	Xiaoju Zhang Abbott Laboratories No competing interests.
REVIEW RETURNED	18-Aug-2017

GENERAL COMMENTS	The authors had revised the previous manuscript, and following related issues need to be addressed. 1). The revised version manuscript removed the criterion "MFS developed within 6 months before cancer" (Figure 1), however, comparing to the previous manuscript, Marfan syndrome recorded is 4 patients less (Table 1). I have trouble to understand that. 2). What does "*" in Table 4 indicate? "P<0.05" statement is not clear to me.
---

VERSION 2 – AUTHOR RESPONSE

Reviewer: 3

Reviewer Name: Giuseppe Biondi-Zoccai

Institution and Country: Sapienza University of Rome, Latina, Italy

Please state any competing interests or state 'None declared': None declared

Please leave your comments for the authors below

Comment: All my comments have been satisfactorily addressed.

Response: Thank you for your valuable comments.

Reviewer: 4

Reviewer Name: Xiaoju Zhang

Institution and Country: Abbott Laboratories

Please state any competing interests or state 'None declared': No competing interests.

Please leave your comments for the authors below

The authors had revised the previous manuscript, and following related issues need to be addressed.

Comment 1. The revised version manuscript removed the criterion "MFS developed within 6 months before cancer" (Figure 1), however, comparing to the previous manuscript, Marfan syndrome recorded is 4 patients less (Table 1). I have trouble to understand that.

Response: Thank you for your comments. In previous revision, a reviewer had raised the concern regarding patients with MFS were more likely to be in routine medical follow-up than an average person of the same age, and therefore more likely to have a malignancy diagnosed. Therefore, in addition to remove criterion "MFS developed within 6 months before cancer", we have reanalyzed the data with matching the case and control groups on the number of medical follow-ups, in addition to age, gender and comorbidities. We confirm the change in the number in the "Patients without malignancy" group (from 31 to 27 MFS patients) was due to matching the number of medical follow up additionally. Please see page 8, line 6 and page 10, line 6.

Comment 2. What does "*" in Table 4 indicate? "P<0.05" statement is not clear to me.

Response: Thank you for your reminding. The asterisk is unnecessary. We have removed the mark. Please see page 14, table 4.

Reviewer: 2

Reviewer Name: Hazem Elsebaie

Institution and Country: Cairo University, Egypt

Please state any competing interests or state 'None declared': None declared

Please leave your comments for the authors below

Comment: The author's reply, clarifications and amendments seem satisfactory.

Response: Thank you for your valuable comments.

VERSION 3 – REVIEW

REVIEWER	Xiaoju Zhang Abbott Laboratories, USA No conflict of interest.
REVIEW RETURNED	22-Sep-2017
GENERAL COMMENTS	The revisions meet the reviewer's expectation. Previous addressed concerns are responded accordingly. Hence, the reviewer agree to accept the manuscript.